# Online Explanation Generation for Human-Robot Teaming

Mehrdad Zakershahrak, Ze Gong, Nikhillesh Sadassivam, Akkamahadevi Hanni and Yu Zhang[1]

*Abstract*— As Artificial Intelligence (AI) becomes an integral part of our life, the development of explainable AI, embodied in the decision-making process of an AI or robotic agent, becomes imperative. For a robotic teammate, the ability to generate explanations to explain its behavior is one of the key requirements of an explainable agency. Prior work on explanation generation focuses on supporting the reasoning behind the robot's behavior. These approaches, however, fail to consider the mental workload needed to understand the received explanation. In other words, the human teammate is expected to understand any explanation provided, often before the task execution, no matter how much information is presented in the explanation. In this work, we argue that an explanation, especially complex ones, should be made in an online fashion during the execution, which helps spread out the information to be explained and thus reducing the mental workload of humans. However, a challenge here is that the different parts of an explanation are dependent on each other, which must be taken into account when generating online explanations. To this end, a general formulation of online explanation generation is presented along with three different implementations satisfying different online properties. We base our explanation generation method on a model reconciliation setting introduced in our prior work. Our approaches are evaluated both with human subjects in a standard planning competition (IPC) domain, using NASA Task Load Index (TLX), as well as in simulation with ten different problems across two IPC domains.

## I. INTRODUCTION

As intelligent robots become more prevalent in our lives, the interaction of these AI agents with humans becomes more frequent and essential. One of the most important aspects of human-AI interaction is for the AI agent to provide explanations to convey the reasoning behind the robot's decision-making [1]. An explanation provides justifications for the agent's intent, which helps the human maintain trust of the robotic peer as well as a shared situation awareness [2], [3]. Prior work on explanation generation often focuses on supporting the motivation for the agent's decision while ignoring the underlying requirements of the recipient to understand the explanation [4], [5], [6]. However, a good explanation should be generated in a lucid fashion from the recipient's perspective [7].

To address this challenge, the agent should consider the discrepancies between the human and its own model while generating explanations. In our prior work [7], we encapsulate such inconsistencies as *model differences*. An explanation then becomes a request to the human to adjust

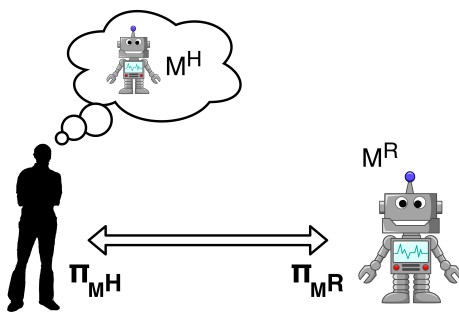

Fig. 1: The model reconciliation setting [7]. $M^R$ represents the robot's model and $M^H$ represents the human's model of expectation. Using $M^H$, the human generates $\pi_{M^H}$, which captures the human's expectation of the robot. Whenever the two plans are different, the robot should explain by generating an explanation to reconcile the two models.

the model differences in his mind so that the robot's behavior would make sense in the updated model, which is used to produce the human's expectation of the robot. The general decision-making process of an agent in the presence of such model differences is termed *model reconciliation* [7], [8].

One remaining issue, however, is the ignorance of the mental workload required of the human for understanding an explanation. In most earlier work on explanation generation, the human is expected to understand any explanation provided regardless of how much information is present and no discussion has been provided on the process for presenting the information. In this work, we argue that explanations, especially complex ones, should be provided in an online fashion, which intertwines the communication of explanations with plan execution. In such a manner, an online explanation requires less mental workload at any specific point of time. One of the main challenges here, however, is that the different parts of an explanation could be dependent on each other, which must be taken into account when generating online explanations. The online explanation generation process spreads out the information to be communicated while ensuring that they do not introduce cognitive dissonance so that the different parts of the information are perceived in a smooth fashion.

### A. Motivating Example

Let us illustrate the concept of online explanations through a familiar situation between two friends. Mark and Emma want to meet up to study together for an upcoming exam. Mark is a take-it-easy person so he plans to break the review session into two 60 minutes parts, grab lunch in between the sub-sessions and go for a walk after lunch. On the other

[1]Mehrdad Zakershahrak, Ze Gong, Nikhillesh Sadassivam, Akkamahadevi Hanni and Yu Zhang are with the School of Computing, Informatics and Decision Systems Engineering, Arizona State University, Tempe, AZ. {mzakersh,zgong11,nsadassi, ahanni,Yu.Zhang.442}@asu.edu

hand, Mark knows that Emma is of a more focused type who would rather keep the review in one session and get lunch afterwards. Mark would like to keep his plan. However, had he explained to Emma at the beginning of his plan, he knew that Emma would have proposed to order takeouts for lunch on the way before the review session. Instead, without revealing his plan, he goes with Emma to the library. After studying for 60 minutes, he then explains to Emma that he cannot continue without energy, which makes going to lunch the best option for both. At the same time, Mark refrained from telling Emma (until after lunch) that he also needed a walk since otherwise Emma would have proposed for him to take a walk alone while she stays a bit longer for review, and then to meet up at the lunch place.

The above example demonstrates the importance of providing an explanation in an online fashion. Mark gradually reveals the reasoning to maintain his plan as the execution unfolds so that it also becomes both acceptable and understandable to Emma, even though being subject to different values due to model differences (e.g., Mark values lunch break more than Emma thinks he does). The key point here is to explain minimally and only when necessary. In this way, the information to be conveyed is spread out throughout the plan execution, potentially with even a reduced amount of information, so that there is less mental workload requirement at the current step–from Emma's perspective, the interaction with Mark is more straightforward.

In this paper, we develop a new method for explanation generation that intertwines explanation with plan execution. The new form of explanation is referred to as *online explanation*, which considers the mental workload of the receiver of an explanation by breaking it into multiple parts that are to be communicated at different times during the plan execution. We implemented three different approaches for online explanation generation, each focusing on different "online" properties. In the first approach, our focus is on matching the plan prefix. In the second approach, the focus is on making the very next action understandable to the human teammate. In the third approach, the focus is on matching the prefix of the robot's plan with any possible optimal human plan. We use a model search method that ensures that the earlier information communicated would not affect the later parts of the explanation. This creates a desirable experience for the recipient by significantly reducing the mental workload. Our approaches are evaluated both with human subjects and in simulation.

## II. RELATED WORK

AI and its numerous applications have provided astounding benefits in areas such as transportation, medicine, finance and military in recent years, but AI agents are so far limited in their ability to operate as a teammate. To be considered a teammate, the agent must not only achieve a given task, but also provide a level of transparency to other members of the team [3]. One of the ways to achieve this is to enable AI agents to be self-explanatory in their behaviors. Recently, explainable AI paradigm [9] rises as one essential constituent

of human-AI collaboration. Explainable AI helps improve human trust of the AI agent and maintain a shared situation awareness by contributing to the human's understanding of the underlying decision-making process of the agent.

The explainable agency's effectiveness [10] is assessed based on its capability to model the human's perception of the AI agent accurately. This means that an explainable AI agent must not only model the world, but also the other agents' perception of itself [11]. This model of the other agents allows the agent to infer about their expectation of itself. Using this model, an agent can generate legible motions [12], explicable plans [8], [13], [14], or assistive actions [15]. In these approaches, an agent often substitutes cost optimality with a new metric that simultaneously considers cost and explicability. Another way of using the model is for an AI agent to signal its intention before execution [16]. The motivation here is to use the model to search for additional context information that would help improve human understanding.

A third way of using this model is for the agent to explain its behavior by generating explanations [4], [5], [6]. Similar to intention signaling, this method has the benefit that the agent can maintain its optimal behavior. Research along this direction has focused on generating the "right" explanations based on the recipient's perception model of an explanation [7], [17]. This is useful, however, only with the assumption that the explanation can be understood, regardless of how much information is provided or whether sufficient time is given–the mental workload that is required for understanding an explanation is largely ignored.

In our prior work, we have studied how the ordering of the information of an explanation may influence the perception of an explanation [18]. In this work, we further argue that an explanation must sometimes be made in an online fashion. This is especially true for complex explanations that require a large amount of information to be conveyed. The idea behind online explanation generation is to provide a minimal amount of information that is sufficient to explain part of the plan that is of interest currently (e.g., the next action), and in such a way intertwine explanation generation with plan execution.

## III. EXPLANATION GENERATION

Our problem definition is based on the model reconciliation setting defined in our prior work [7]. We provide a brief review of the relevant concepts before defining our problem in this work. Our problem is closely associated with planning problems so we first provide the background here. A planning problem is defined as a tuple $(F, A, \mathcal{I}, \mathcal{G})$ using PDDL [19], similar to STRIPS [20]. $F$ is the set of predicates used to specify the state of the world and $A$ is the set of actions used to change the state of the world. Actions are defined with a set of preconditions, add and delete effects. $\mathcal{I}, \mathcal{G}$ are the initial and goal state.

*Definition 1 (Model Reconciliation):* : A model reconciliation is a tuple $(\pi^*_{I,G}, \langle M^R, M^H \rangle)$, where $cost(\pi^*_{I,G}, M^R) = cost^*_{M^R}(I, G)$ and $\pi^*_{I,G}$ is the robot's plan to be explained.

Where $cost(\pi_{I,G}^*, M^R)$ is the cost of the plan generated using $M^R$ and $cost_{M^R}^*(I,G)$ is the cost of the optimal plan based on the initial and goal state pair under $M^R$. In other words, the robot plan to be explained is required to be optimal according to $M^R$, assuming rational agents. The model reconciliation setting also takes the human's model $M^H$ into account, which captures the human's expectation of the robot's behavior. When the robot's behavior to be explained (i.e., $\pi_{I,G}^*$) matches with the human's expected behavior, the models are said to be reconciled for the plan. A figure that illustrates the model reconciliation setting is presented in Figure 1. Explanation generation in a model reconciliation setting means bringing two models, $M^H$ and $M^R$, *"close enough"* by updating $M^H$ such that $\pi_{I,G}^*$, the robot's plan, becomes fully explainable (optimal) in the human's model. A mapping function was defined in [7] to convert a planning problem into a set of features that specifies the problem as $\Gamma: \mathcal{M} \longmapsto S'$ is a mapping function, which transfers any planning problem $(F, A, \mathcal{I}, \mathcal{G})$ to a state $s'$ in the feature space as follows:

$$
\tau(f) = \begin{cases} init - has - f, & \text{if } f \in \mathcal{I}. \\ goal - has - f, & \text{if } f \in \mathcal{G}. \\ a - has - precondition - f, & \text{if } f \in pre(a), a \in A. \\ a - has - add - effect - f, & \text{if } f \in eff^+(a), a \in A. \\ a - has - del - effect - f, & \text{if } f \in eff^-(a), a \in A. \\ a - has - cost - f, & \text{if } f = c_a, a \in A. \end{cases}
$$

$$
\Gamma(\mathcal{M}) = \{\tau(f) | \forall f \in \mathcal{I} \cup \mathcal{G} \cup \bigcup_{a \in \mathcal{A}} \{f' | \forall f' \in \{c_a\} \cup pre(a) \cup eff^+(a) \cup eff^-(a)\}\}
$$

In other words, the mapping function converts a planning problem into a set of features that specifies the problem.

*Definition 2 (Explanation Generation [7]):* The explanation generation problem is a tuple $(\pi_{I,G}^*, \langle M^R, M^H \rangle)$, and an explanation is a set of unit feature changes to $M^H$ such that 1) $\Gamma(\widehat{M^H}) \setminus \Gamma(M^H) \subseteq \Gamma(M^R)$, and 2) $cost(\pi_{I,G}^*, \widehat{M^H}) - cost_{\widehat{M^H}}^*(I,G) < cost(\pi_{I,G}^*, M^H) - cost_{M^H}^*(I,G)$, where $\widehat{M^H}$ is the model after the changes.

An explanation hence reconciles two models by making the cost difference between the human's expected plan and the robot's plan smaller after the model updates.

*Definition 3 (Complete Explanation [7]):* Given an explanation generation problem, a complete explanation is an explanation that satisfies $cost(\pi_{I,G}^*, \widehat{M^H}) = cost_{\widehat{M^H}}^*(I,G)$.

The robot's plan must be optimal in the human's model after a complete explanation ($\widehat{M^H}$). A minimal complete explanation (MCE) [7] is defined as a complete explanation that contains the minimum number of unit feature changes.

## IV. ONLINE EXPLANATION GENERATION (OEG)

While the previous explanation generation approach provides a framework to generate explanations considering both the robot's model and the human's model, it largely ignores the mental workload requirement of the human for understanding the explanation. We introduce *online explanation*

*generation* to address this issue. The key here is to only provide a minimal amount of information during the plan execution to explain the part of the plan that is of interest and not explainable.

*Definition 4 (Online Explanation Generation):* Given a model reconciliation problem, an online explanation is a set of sub-explanations $(e_k, t_k)$, where $e_k$ represents the $k$th set of unit features to be made (as a sub-explanation) at step $t_k$ in the plan.

Basically, an online explanation requires only that any actions in the robot's plan before the $k$th sub-explanations will match with that of the human's expectation. In such a way, the robot can split an explanation into multiple parts, which are made in an online fashion as the plan is being executed. We provide three different approaches of online explanation generation based on the definition provided, while each of these approaches focus on one aspect of explanation generation intertwined with plan execution. Section IV-A discusses *OEG with Plan Prefix matching*, Section IV-B describes *OEG with Next Action matching* and Section IV-C explains *OEG with any prefix matching*.

### A. OEG for matching Plan Prefix (OEG-PP)

To generate the sub-explanations (i.e., $\{e_k\}$) for an online explanation, the planning process must consider how the sequence of model changes would result in the changes of the human's expectations after each sub-explanation. Similar to the search process for complete explanations [7], we convert the problem of explanation generation to the problem of model search in the space of possible models. The challenge here is that the model changes may not be independent, i.e., future changes may render a mismatch in the previously reconciled plan prefixes. To address this issue, it must be ensured that the model changes after $e_k$, i.e., $e_{k+1:m}$ where $m$ denotes the size of the set of sub-explanations, would not change the plan prefixes in $M^H$. This can be achieved by searching from $M^R$ to $M^H$ to find the largest set of model changes which ensure that the plan prefix would not change afterwards after further sub-explanations. This search process is illustrated in Figure 2. An OEG-PP is a set of sub-explanations $(e_k, t_k)$ such that:

$$
\forall k > 1, Prefix(\pi_{I,G}^*, t_k - 1) = Prefix(\pi_{E_{k-1}}^H, t_k - 1)
$$
$$
\Gamma(M_{E_{k-1}}^H) = \Gamma(M^H) \cup E_{k-1}
$$
$$
s.t.
$$
$$
\bigcup_i e_i = \Gamma(\widehat{M^H}) \setminus \Gamma(M^H) \subseteq \Gamma(M^R)
$$
(1)

where $Prefix(\pi, t)$ returns the prefix of a plan $\pi$ up to step $t_{k-1}$. $E_k$ represents $e_{1:k}$ and $\pi_{E_k}^H$ is the optimal plan created from $M_{E_k}^H$ ($M^H$ after providing sub-explanations $e_1$ to $e_k$). More specifically, the following process will be performed recursively for each sub-explanation. First, we continue moving along $\pi_{I,G}^* = (a_1, a_2, ..., a_n)$ as long as the plan prefix matches with the prefix of the plan using the human model $M^H$. Let $t = t_1$ be the first plan step where they differ. Our search for the sub-explanation starts

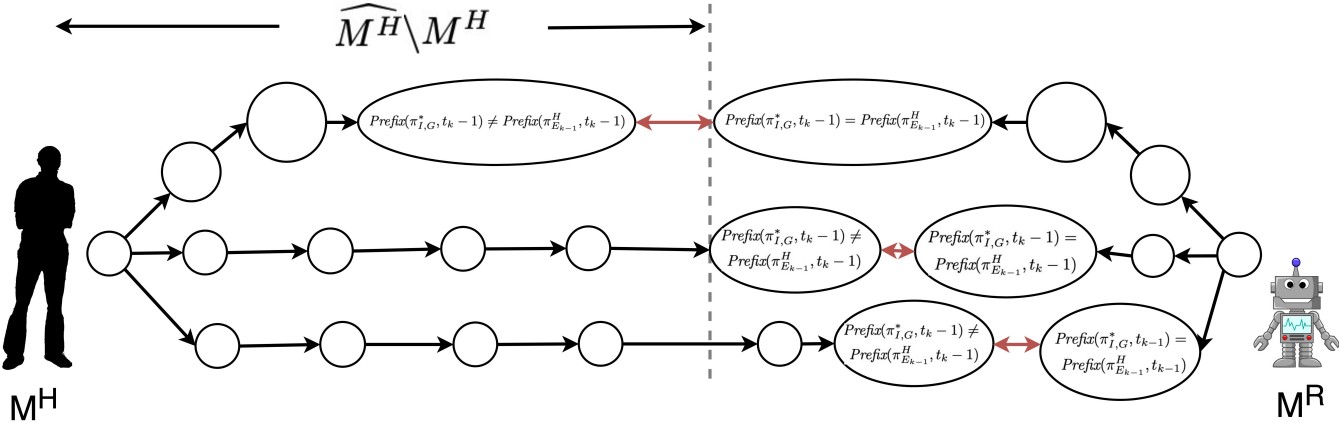

Fig. 2: Model space search process for OEG-PP. Compared to MCE in the previous work [7], the difference is that in our approach the search starts from the robot model and stops where the plan prefixes for the updated human model and the robot model match, while in the previous approach the search process starts from the human model ($M^H$). In this aspect, our research process is more akin to MME [7]. However, since we are focusing on matching the prefixes rather than the whole plan in one shot, our approach must run this process multiple times compared to only once in MME. While seemingly more computationally expensive, this characteristic actually allows us to beat both MCE and MME in terms of computation since our approaches at any time consider only a small set of changes (see results). The dotted line represents the border of the maximum state space model modification in robot model which reconciles the two models up to where the plan execution currently is. Maximum updates to the robot model is equivalent to minimum updates to the human model.

with $M^R$. It finds the largest set of model changes to $M^R$ such that the prefix of a plan using the corresponding model (i.e., $M^R$ minus the set of changes) matches with that of $\pi^*_{I,G}$ up to step $t_2 - 1$. The complement set of changes (i.e., the difference between $M^H$ and $M^R$, minus this set of changes) will be $e_1$. For the next recursive step, we will start from action $t_1$ and the human model will be $M^H_{E_1}$. To ensure that the prefix (up to $t_2 - 1$) will be maintained for future steps, we directly force the later plans to be compatible with the prefix. Since we know that an optimal plan exists that satisfies this requirement following the search process, this would not affect our solution for online explanation.

The recursive search algorithm for model space OEG is presented in Algorithm 1 for finding $e_k$ given $E_{k-1}$. To search for $e_k$, we use a recursive model reconciliation procedure on the model space. Given $M^H_{E_{k-1}}$ and $M^R$, we start off with finding the difference between these two models, and modify $M^R$ with respect to $M^H$ to find the largest set of model changes that can satisfy constraints introduced in Eq. (1). This algorithm continues until the human's plan matches with that of the robot's plan.

### B. OEG for matching Next Action (OEG-NA)

Throughout OEG-PP, we assume that generating explanations would modify $M^H$, and the goal of explanation generation is to ensure that the robot and human plan have the same prefix at any step of plan execution. However, this is not always required since the human may not be interested in actions that occurred. Hence, we relax earlier than the current action the plan prefix condition, such that the robot needs only to reconcile between $M^R$ and $M^H$ to match the very next action in $\pi^*_{I,G}$ and $\pi^H_{E_{k-1}}$ at step $t_k$, regardless of the earlier actions in the plan prefix. This approach is also motivated by the fact that the human is

---

**Algorithm 1:** OEG-PP Algorithm

> **input** : $M^H_{E_{k-1}}$, $M^R$, $\pi^*_{I,G}$ and $\{e_{k-1}, t_k - 1\}$
> **output:** Sub-explanation $e_k$
> Compute $\Delta(M^H_{E_{k-1}}, M^R)$ as the difference between the two models;
> Sort $\Delta(M^H_{E_{k-1}}, M^R)$ ascending based on the size of the model changes;
> Compute $\pi_H$ based on $M^H_{E_{k-1}}$ with prefix set up to $t_k - 1$;
> $t_k \leftarrow$ FirstDiff($\pi^*_{I,G}, \pi_H$);
> ▷ The first plan difference between $\pi^*_{I,G}$ and $\pi_H$
> LONGESTMONOTONIC($M^H_{E_{k-1}}, t_k, \Delta$)
> **if** ($\pi^*_{I,G} \equiv \pi_H$) **then**
>   | return $\{\}$;
> **for** $\forall f \in \Gamma(M^R) \backslash \Gamma(M^H_{E_{k-1}})$ **do**
>   | ▷ All remaining differences after sub-explanations $E_{k-1}$
>   | $\lambda \leftarrow \Gamma(\overline{M^H_f})$ ;   ▷ create a modification
>   | **if** $\pi^H_{E_{k-1}} \equiv \pi^*_{I,G}$ **then**
>   |   | return $\lambda$;
>   | Create a plan $\pi^f_H$ using $(\overline{M^H_f})$;
>   | **if** ($t_k \leq$ *FirstDiff*($\pi^f_H, \pi^*_{I,G}$)) **then**
>   |   | **if** $|\lambda| > \lambda_{max}$ **then**
>   |   |   | $\lambda_{max} \leftarrow \lambda$;
>   |   |   | $\Delta(M^H_{E_{k-1}}, M^R) -= \lambda_{max}$ ;
>   |   |   | Sort ($\Delta(M^H_{E_{k-1}}, M^R)$) ascending ;
>   |   |   | LONGESTMONOTONIC($M^H_{E_{k-1}}, t_k, \Delta$) ;
>   | return $\lambda_{max}$ as $e_k$;

known to have limited cognitive memory span [21]. In the most limited case, the agent focuses on explaining the very next action that is different between the most recent human plan $\pi_{E_{k-1}}^H$ and $\pi_{I,G}^*$. Similar to Algorithm 1, We perform a recursive model reconciliation procedure on the model space. Compared to other two approaches, first, we perform the search from $\widehat{M^H} \setminus M^H$ rather than $M^R$ (see Figure 2) since it is computationally faster due to the fact that the plan prefixes do not need to be identical and since the search procedure is monotonic, the search result would be equivalent as if the procedure started from $M^R$. The other difference here is that we do not compare the entire plan prefix. Instead, the agent explains only the immediate next action that does not match in the human and robot plans that, without requiring the explanation also maintains the match between the prefixes. In this aspect, the search process of OEG is similar to that of minimally monotonically explanation (MME) in [7], except that the process must be executed multiple times for OEG due to its online fashion. In the implementations, however, our algorithms actually combine search from $M^H$ and $M^R$ for a better performance, given the fact that latter model updates do not often affect the previous sub-explanations:

$$\forall k > 1, \forall t_k - 1 \leq t < t_k, a_k \in \pi_{I,G}^*[t] \ \& \ a_k \in \pi_{E_{k-1}}^H[t]$$
$$\& \ \Gamma(M_{E_{k-1}}^H) = \Gamma(M^H) \cup E_{k-1}$$
$$s.t.$$
$$\bigcup_i e_i = \Gamma(\widehat{M^H}) \setminus \Gamma(M^H) \subseteq \Gamma(M^R)$$
(2)

*C. OEG for matching Any Prefix (OEG-AP)*

One assumption in the OEG-PP approach is that the robot has only right plan. Subsequently, the robot's goal is to reconcile the human's plan with respect to its own plan using model space search. We relax this assumption by assuming that there is a set of optimal plans. In such a setting, the robot does not need to explain as long as there exists a human plan that has the same prefix as the robot's plan earlier than the current action. The goal of OEG here is thus to satisfy the following:

$$\exists \pi_{E_{k-1}}^H \in \Pi_{E_{k-1}}^H$$
$$\forall k > 1, Prefix(\pi_{I,G}^*, t_k - 1) = Prefix(\pi_{E_{k-1}}^H, t_k - 1)$$
$$\Gamma(M_{E_{k-1}}^H) = \Gamma(M^H) \cup E_{k-1} \quad (3)$$

where $\Pi_{E_{k-1}}^H$ is a set of optimal plans generated using $M_{E_{k-1}}^H$, $\pi_{E_{k-1}}^H$ is the human optimal plan generated from $M_{E_{k-1}}^H$ and $\pi_{M_H}^*$ is the human optimal plan generated from the original human model ($M^H$). A straightforward solution to OEG-AP is to generate all human optimal plans and check if any one of them matches with the robot's plan (prefix). This approach however is computationally expensive. Instead, we implemented a compilation approach. To check that a plan prefix $Prefix(\pi_{I,G}^*, t_k - 1)$ in the robot's plan is also a prefix in the human's model, we first compile the problem in the human's model into a new problem such that the robot's plan prefix would always be a prefix of the human's plan.

If the cost of the human's optimal plan in this new domain model is equal to the cost of the human's optimal plan before the compilation, then clearly there exists an optimal plan in the human's model that matches the prefix. Otherwise, we know that an explanation must be made. Hence, the key here is to ensure that a plan prefix is always satisfied in the compiled model.

This is not difficult to achieve. For all $i \geq 1$, such that $a_i, a_{i+1} \in Prefix(\pi_{I,G}^*, t_k - 1)$, where $a_i, a_{i+1}$ are two consecutive actions in $\pi_{I,G}^*$, the compilation can be achieved by adding a predicate $p_i$ to $a_i$ as an effect, which is a prerequisite for $a_{i+1}$. $a_{i+1}$, in its turn deletes $p_i$ and adds $p_{i+1}$ which is a prerequisite for $a_{i+2}$, etc.

To search for $e_k$, we again use a recursive model reconciliation process on the model space, similar to Algorihm 1. Similar to IV-A, we start off with finding the difference between these two models. The main difference in this approach is that after each model update after a sub-explanation, the agent checks if there exists a human optimal plan that has the same plan prefix as the robot's plan up until the next action using the compilation approach described above. This check stops when such a plan does not exist and a new sub-explanations must be identified by model space search. This process continues until an optimal human plan exists that matches the robot's plan. Note however that this does not mean that an optimal planner would necessary return the same plan using the human's model.

## V. EVALUATION

We evaluated our approach for online explanation generation both with human subjects and in simulation for the different approaches introduced above and compared the results with Minimally Complete Explanation (MCE) [7] approach. For simulation, the goal is to see that how online explanation is in general different from MCE in terms of the information needed and computation time. We evaluated our approach on ten different problems across the rover domain and barman domain–two standard IPC domain described below. For both human and simulation evaluations, the differences between $M^H$ and $M^R$ are made by randomly removing preconditions from an arbitrarily chosen set of model features. For human subject study, the aim is to confirm the benefits of online explanation generation. Our hypothesis is as follows:

- *Online explanation generation will reduce mental workload and improve task performance.*

We evaluated our approach with human subjects on a modified rover domain (see Sec. V-D).

*A. Rover Domain*

In this domain, the rover is supposedly on Mars and the goal is to explore the space to take rock and soil samples as well as taking images and communicate the results after analysis to the base station via the lander. In order to take any image, the rover must first calibrate its camera with respect to the target. To sample rock or soil, the robot must have an empty space in its storage. At any point of time, the rover only has enough space to store one sample. In order to take

| Problem | OEG-PP | | OEG-NA | | | OEG-AP | | | MCE | |
|---|---|---|---|---|---|---|---|---|---|---|
| | Explanations | Time | Explanations | Distance | Time | Explanations | Distance | Time | Explanations | Time |
| | | | | | Rover | | | | | |
| P1 | 3 (1.5) | 8.89 | 7 (1.167) | 0.4 | 17.929 | 2 (1) | 0.4 | 6.94 | 3 | 28.91 |
| P2 | 5 (1.67) | 22.32 | 7 (1.4) | 0.105 | 42.568 | 3 (1) | 0.105 | 18.30 | 5 | 150.54 |
| P3 | 6 (1.5) | 18.68 | 8 (1.143) | 0.068 | 21.258 | 3 (1) | 0.068 | 1.64 | 5 | 176.16 |
| P4 | 6 (1.5) | 50.97 | 8 (1.33) | 0.131 | 94.783 | 5 (1.25) | 0.131 | 45.36 | 6 | 314.15 |
| P5 | 5 (1.67) | 54.83 | 8 (1.33) | 0.135 | 106.709 | 3 (2) | 0.135 | 50.36 | 4 | 272.76 |
| | | | | | Barman | | | | | |
| P1 | 5 (1.25) | 43.01 | 5 (1.25) | 0.911 | 59.912 | 2 (1) | 0.943 | 24.37 | 5 | 179.95 |
| P2 | 5 (1) | 36.17 | 5 (1) | 0.995 | 33.032 | 3 (1) | 0.899 | 9.36 | 5 | 38.89 |
| P3 | 5 (1.25) | 36.83 | 5 (1) | 0.895 | 46.775 | 3 (1.5) | 0.705 | 9.67 | 5 | 51.84 |
| P4 | 5 (1.25) | 78.42 | 5 (1) | 0.838 | 69.016 | 4 (1) | 0.556 | 20.42 | 5 | 61.86 |
| P5 | 5 (1.67) | 41.88 | 5 (1) | 0.892 | 54.708 | 3 (2) | 0.556 | 10.15 | 5 | 61.48 |

TABLE I: Comparison of number of generated explanations and computation time using different approaches for IPC Rover and Barman Domains.

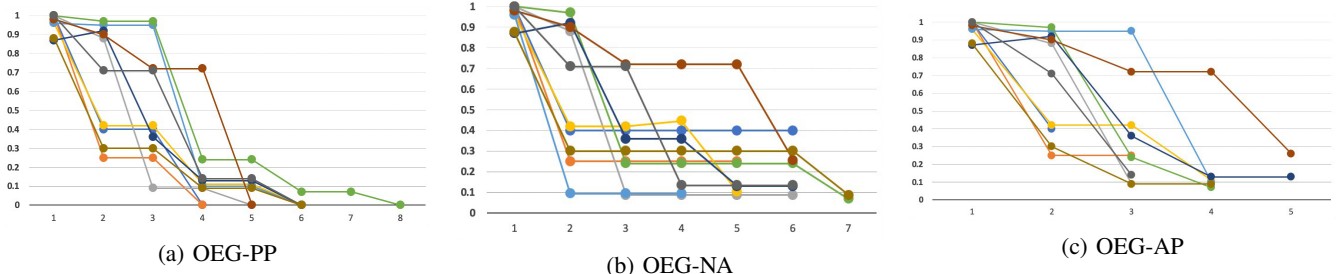

(a) OEG-PP     (b) OEG-NA     (c) OEG-AP

Fig. 3: Plan distance [22] convergence across three different approaches between $\pi^H_{E_{k-1}}$ and $\pi^R$ for the Rover domain problems. The y-axis represents the distances while x-axis represents the number of $E_k$(sub-explanations).

multiple samples, it must drop the current sample before taking another sample [23].

### B. Barman Domain

In this domain, the robot assumes the role of a barman whose goal is to serve a desired set of drinks using drink dispensers, glasses and a shaker. The constraints are that the robot can grab one object if its hand is empty, the robot can grab one object with one hand, and before filling it with a drink, a glass should be empty and clean [23].

### C. Simulation Results

Table I shows the simulation results comparing minimally complete explanations (MCE) withx OEG-PP, OEG-NA and OEG-AP approaches for 5 problems in the rover domain and 5 problems in the barman domain. While the average number of model features of OEG (in a sub-explanation) being shared at each instance of time is considerably lower that MCE (every feature in the explanation is presented at once), the total number of model features in an explanation are the same for MCE and OEG-PP across most of the problems. We can see that in some cases (for instance, P3 from the Rover domain), the total number of model features in the explanation for OEG-PP and OEG-NA is more than that of MCE, which is expected since OEG is focused on generating the minimal amount of information at

each time step, instead of the amount overall. The reason for sharing more information in total in OEG-PP and OEG-NA, when compared to MCE, lies in the dependence between the features and the behavior of the planner (i.e., which optimal is returned). While OEG-AP seems to have improved over the amount of information in an explanation, it actually only shows the advantage of considering all optimal plans instead of the one returned by the planner.

Comparing both the OEG-NA and OEG-AP approaches with MCE and OEG-PP, there is a remaining distance between the robot's plan and the human's plan in terms of plan action distance (also returned by an optimal planner). The distance of OEG-NA is due to the fact that only the immediate next action is considered. For OEG-AP, as we explained, there is no guarantee that the plan returned using the human's model will be the same as the robot's plan since it considers all optimal human plans and only requires one of them to match the robot's. This is also illustrated more clearly in Fig. 3. Furthermore, ion OEG approaches, since the execution and explanation is intertwined, the plan distance [22] between $\pi^H_{E_{k-1}}$ and $\pi^R$ in our approaches gradually moves towards 0 as shown, which suggests a "smoother" adjustment for $M^H$ during the execution. This is expected to have a positive effect on the human's mental workload, which we evaluated next.

Table I also presents the time comparison between differ-

ent approaches. For computation time, the results are collected using a 2015 Mac book Pro, with 2.2 GHz Intel Core i7 and 16 GB of memory. The results of the time comparison suggest that OEG-PP is faster than MCE. Moreover, OEG-NA seems the slowest while OEG-AP is the fastest since it uses fewer model features. The performance improvement over MCE may be surprising, thanks to combining search from $M^R$ and $M^H$. In our implementation, the possible model updates are sorted ascending based on their feature size and our algorithms start checking the ones with the smallest changes from the robot's side. The consistency check is left as we proceed to the next sub-explanation and backtracking is performed when it fails. This search process takes advantage of the fact that latter information often does not affect the previous sub-explanations.

### D. Human Study

To test our hypothesis, we designed a human study to compare our three approaches for online explanation generation with minimally complete explanation (MCE) [7]. Furthermore, to ensure that the performance difference is not solely due to simply breaking information into multiple pieces, we also implement another approach that randomly breaks MCE during plan execution (referred to as MCE-R). We conducted our experiment using Amazon Mechanical Turk (MTurk) with 3D simulation. The subjects were given an introduction to the rover domain and the task they were supposed to help with. Each subject was given a 30-minute limit to finish the task. Explanations were provided using plain English language and rover actions were depicted using GIF images from a 3D simulated scenario as the rover executes the plan. Figure 4 shows the 3D simulated scenario presented to the subjects. In this experiment, the human subject acts as the rover's commander, where the robot is on Mars and supposed to perform a mission autonomously. The human subject observes the rover's plan sequentially and is asked to determine whether the rover's current action is questionable or not, with explanations provided by OEG approaches or MCEs. Each subject can only perform the task for one setting to reduce the influence between different runs. To observe the effect on mental workload more clearly, we have also added a few spatial puzzles to the experiment as a secondary task to create additional cognitive demand.

In the scenario, we deliberately remove certain information from the domain so that the subject would create an incorrect plan, when no explanation is given. In particular, we did not inform them that the storage is limited, the memory is limited, the camera must be calibrated, and the camera must be calibrated with respect to the objective. This hidden information introduces differences between $M^H$ and $M^R$ in the model reconciliation setting, and hence resulting in scenarios where explanations must be provided. In this scenario, for example, the subject may question the action for calibrating the camera if they were not specifically told to consider that.

In MCE setting, the robot shares all the information at the beginning of the task [7], while the information is randomly broken to be communicated at different steps in MCE-R. In each of the OEG setting, the robot uses different approaches of online explanation generation, which intertwines the communication of explanation with the plan execution. In particular, the four pieces of missing information are provided to the subjects at different steps. In all settings, the subjects were asked to determine whether the robot's action makes sense or not at a time. The minimally complete explanations are generated based on [7] and online explanations are generated using approaches introduced above.

At the end of the study, the subjects were provided the NASA Task Load standard questionnaire to evaluate the efficiency of different explanation approaches by NASA Task Load Index (TLX) [24]. The NASA TLX is a subjective workload assessment tool to evaluate human-machine interface systems. Mental workload is a multidimensional variable which can be captured by different variables and NASA TLX is one of the most frequently used subjective measurements for capturing different aspect of mental workload [25]. It calculates an overall mental workload score using a weighted average on sub-scales: mental demand, physical demand, temporal demand, performance, effort and frustration. Since our experiment does not involve physical demand, we did not include the corresponding question. The description of questions used for each category is presented as follows:

- *Mental Demand*: How mentally demanding was the task?
- *Temporal Demand*: How hurried or rushed was the pace of the task?
- *Performance*: How successful were you in accomplishing what you were asked to do?
- *Effort*: How hard did you have to work to accomplish your level of performance?
- *Frustration*: How insecure, discouraged, irritated, stressed, and annoyed were you?

### E. Human Study Results

We created the academic survey using Qualtrics and recruited 150 human subjects on MTurk, 30 subjects for each setting. To improve the quality of the responses, we set the criteria that the worker's HIT acceptance rate must be greater than 98%. After sifting out invalid responses (i.e., failing to identify the two purposely inserted random actions), we had 94 valid responses in total: 19 for each of MCE-R and MCE, 20 for OEG-PP, and 18 for each of OEG-NA and OEG-AP. The age range of subjects was between 18 and 70, and 29.8% of the subjects were female.

We examined how well the human subjects understand the robot's plan given the different explanations, and compared the distances across the five different settings. We compute the distance between the robot's plan and the human's expected plan by the ratio between the number of questionable actions and the total number of actions in a plan. The lower the distance value, the closer the human's plan is to the robot's plan. This metric intuitively captures how much the human subject understands the robot's plan. We calculated the averaged results of each settings over all of the subjects participated in that setting, using subjective questions from

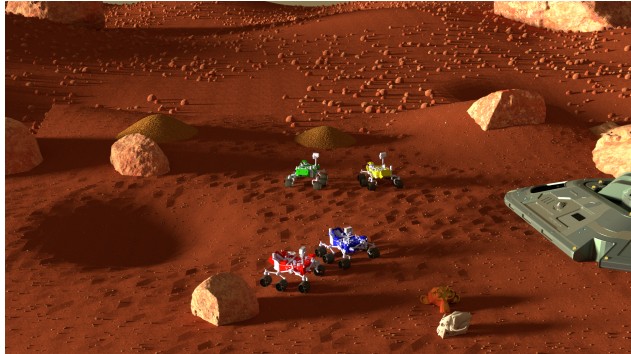

(a) The blue rover moves between waypoints

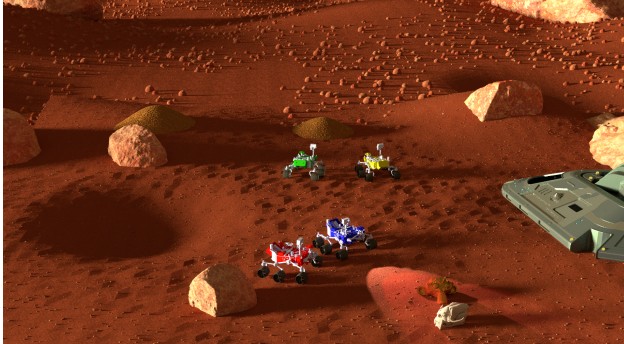

(b) The blue rover takes a picture of one of the objectives

Fig. 4: The 3D visualization of the modified IPC rover domain problem provided to the human subjects. The rovers must together take pictures of targets, collect rock and soil samples, and transmit them to the lander after analysis. The subject views the actions of the rovers via GIF images. (a) and (b) shows the begin and end of an action in which one of the rover takes an image of a target at the bottom.

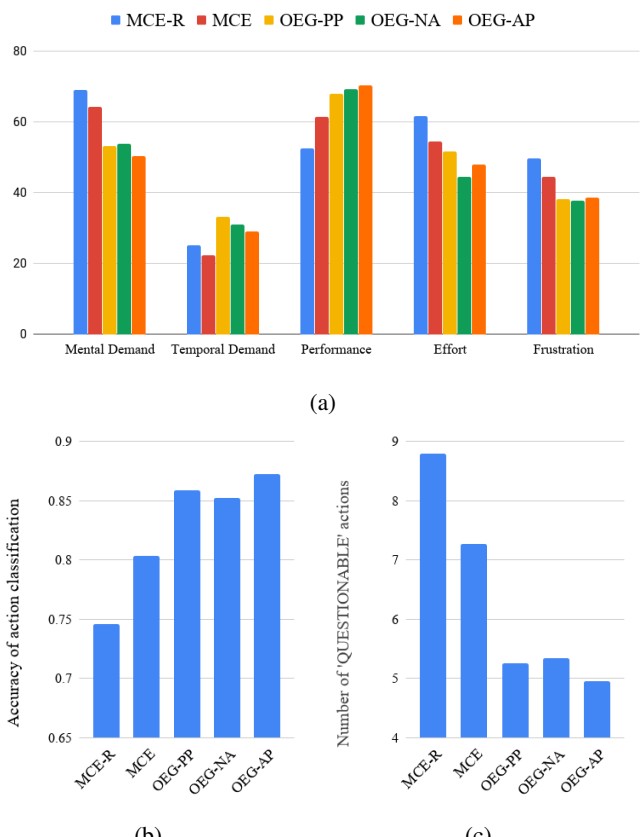

Fig. 5: (a) Comparison of the results of all of TLX categories for the five settings (b) Accuracy of action classification (c) Number of questionable actions

NASA TLX and objective performance measures such as the number of questionable actions and the accuracy of action classification. Results are shown in Figure 5.

The results overall show that OEG approaches are able to better reduce the human's mental workload than MCE approaches. This is backed up by the fact that OEG approaches resulted in better performance in almost all NASA TLX measures. Due to intertwining the explanation process with the plan execution, the OEG approaches create more temporal demand according to the experiment, which is expected. Figure 5 presents both objective performance measures, and subjective results of the human study amongst the 5 TLX categories. First, the number of questionable actions are significantly lower among OEG approaches when comparing to the MCEs. This indicates that the subjects had more trust towards robots in the OEG cases. Moreover, the accuracy of identifying the correct actions (questionable vs. non-questionable) among OEG approaches are higher. Between the three approaches, OEG-AP has the least number of questionable actions and the most accuracy.

We have also presented the p-value for the mental load based on the subjective measures in Table 6 (with weights 1 for all measures ranging from 0 to 100). The results indicate a statistical significant difference between OEG approaches and MCEs for the mental workload in a pairwise comparison. The overall p-value across five categories is $0.0068$ between OEGs (as a group) and MCEs (as a group).

We also did some time analysis. The average overall time taken to accomplish the task for each of the categories is as follows: OEG-NA (567.44s) < OEG-AP (629.56s) < MCE-R (678.98s) < MCE (763.47s) < OEG-PP (775.65s), although we did not see a statistically significant difference due to large variances. The accuracy of the secondary task is also not significantly different between the various approaches.

## VI. CONCLUSION

In this paper, we introduced a novel approach for explanation generation to reduce the mental workload needed for the human to interpret the explanations, throughout a human-robot interaction scheme. The key idea here is to break down a complex explanation into smaller parts and convey them in an online fashion, while intertwined with the plan execution. We take a step further from our prior work by considering not only providing the correct explanations, but also the explanations that are easily understandable. We provided three different approaches each of which focuses on one aspect of explanation generation weaved in plan execution. This

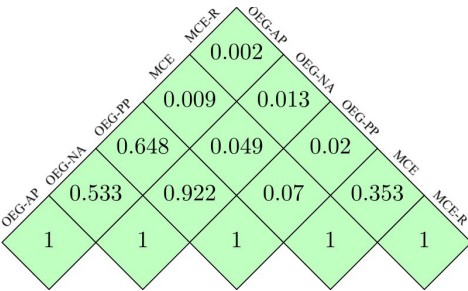

Fig. 6: p-values across different approaches on the mental workload, which is the sum of subjective measures (with weights 1).

is an important step toward achieving explainable AI. We evaluated our approaches using both simulation and human subjects. Results showed that our approaches achieved better task performance while reducing the mental workload.

## ACKNOWLEDGMENT

This research is supported in part by the NSF grant IIS-1844524, the NASA grant NNX17AD06G, and the AFOSR grant FA9550-18-1-0067.

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
