# OpenReview forum: "Online Explanation Generation for Human-Robot Teaming"
_icaps-conference.org/ICAPS/2019/Workshop/XAIP — XAIP 2019_

### Official Review · AnonReviewer1 · 2019-05-08
**Interesting and relevant approach to explanation, but motivation and description of evaluation weak**

**Rating:** 2
**Confidence:** 2

**Review:**

The paper presents an approach to generating explanations online through model reconciliation. The smallest reconciliation that explains the plan prefix is given. This allows the user to understand the completed behavior, without becoming overloaded by a large explanation.

The approach is straight-forwardly explained and interesting. The main weakness of the paper is the motivation, which is not very clear from the motivating example alone. The evaluation setup is not fully explained, but could serve to better explain how the approach is intended to aid the user. Such an example, showing how the user interacts with the plan and the system, highlighting the problems with offline explanation, would definitely strengthen the paper.

About the motivating example: rather than explain the plan, and increase the Emma's understanding about what is going to happen, the approach instead seems to hide information from Emma until it is impossible for her to backtrack and alter the plan. I feel that in this example, the lack of clarity on what is planned should lead to more confusion and chance for failure, rather than less. For example, why did Emma not buy food on the way to the study session without telling Mark? Although she did not have an explicit question about the plan, it was clear (to Mark) that she had an assumption that he knew to be false. The motivations for the approach is stated to be that the explanation is more easily understood if broken into multiple parts. The reason for bringing explanations online in the example is instead that the plan becomes acceptable (rather than understandable).

It is not clear to me from this example why online explanations reduce the frustration and mental demand of Emma. I'm not convinced (by this example) that offline explanations are worse. In fact, the same approach described in this paper could be applied as offline explanation, in which the explanation is given in parts to the user in order to reduce mental demand, without introducing any temporal demand or early commitment. Can you provide some examples in which online explanation is necessary for the user to understand a plan (before it is already executed)?

The algorithm is not completely clear to me. My understanding from the text is a set of changes reconciling M^R and M^H is found (\lambda). Then, the maximum set of f\in\lambda is found such that the plan prefixes of M^R and M^R minus the maximum set remain the same. Then, the complement of this set is the minimum set of changes required to explain the prefix.
- The set is found by breadth-first search, however algorithm 1 appears to add the changes to lambda in order and inside a single loop until the prefix remains unchanged. Which is it?
- Algorithm 1 also returns the set \lamdba_max as e_k, where instead should it not return the complement of this set?

With only 11/14 responses, it is hard to tell if there is a significant difference in any of the results shown in figure 3. The authors do state that no significant difference was observed due to insufficient data points collected. However, despite this, I would be interested to see more extensive results from this same experiment.
Given this:
- How was the experiment carried out? Where the users controlling the rover, or only observing the rover? If they had some control, how was their plan and the AI plan used in the simulation?
- Was there any temporal element to the simulation? If so, how long did actions last? Where there any time limits?
- With MCE, how long (offline) were users given to understand the explanation?

Expanding more on the last two points. It seems to me that the drawback to online explanation that is not mentioned in the paper is the additional time it takes to complete online. In contrast, a large offline explanation might be difficult to parse completely, but time can be devoted to understanding the explanation before confirming any plan.

- Taking the explanations online much reduces the time available to parse the explanation. A fair test would be to compare online explanations against MCE, in which the user is given offline time to understand the explanation. The amount of time allowed offline depends upon the domain. Perhaps a motivating example would be a domain/scenario in which there is little time for a human user to understand and adjust plans before beginning execution?

- It appears that the user does not have a chance to confirm the plan and the purpose of the explanation is to justify the robot's already completed behavior, not to confer understanding the long-term effects of current choices. It would be more interesting to see a domain in which there is the possibility of dead-ends and failure, or plan cost. If the user is working with the robot, then an effective explanation should allow the user to obtain a better cost, or complete the task more reliably.

- The discussion of the evaluation focuses on the mental demand column, but does not mention the additional temporal demand or additional effort. I would say that the difference is also not significant, but that the summary in the abstract and conclusions are a bit one-sided.

---

### Official Review · AnonReviewer2 · 2019-05-09
**Interesting Foray into Online Explanations along with a nice tentative study of HSR - though results very limited in that regard.**

**Rating:** 5
**Confidence:** 2

**Review:**

Motivating Example: Interesting, this gets into some Theory of Mind stuff to me, but I'm curious how it gets addressed later in the paper. "Online Model Reconciliation" maybe?

Figure 1: the text in the bubbles is too small to read. I therefore lose the point of seeing it, though it does look nice.

Conclusion: What next?

Minor:

  Definition 1: the gap between the end of this and the start of the next paragraph looks like its somehow formatted oddly? Rearranging that sentence of cost(\pi,M) so that it's not so close to the above italics would probably fix it.

  Definition 2: Your equations mess w/ the padding of the line. I might break it out similar to your definition of r(f) above.

---

### Decision · Program_Chairs · 2019-05-15

**Decision:**

Accept

**Comment:**

While the reviewers do not fully agree on the decision, in the spirit of making the workshop a venue for discussion and feedback we decided to reject only those papers with strong reject votes.

Please address all review criticism as best possible for the final paper version and its presentation at the workshop. Looking forward to discuss your work at the workshop!